# Our Experience over 20 Years: Antimicrobial Peptides against Gram Positives, Gram Negatives, and Fungi

**DOI:** 10.3390/pharmaceutics15010040

**Published:** 2022-12-22

**Authors:** Giulio Rizzetto, Daisy Gambini, Andrea Maurizi, Matteo Candelora, Elisa Molinelli, Oscar Cirioni, Lucia Brescini, Andrea Giacometti, Annamaria Offidani, Oriana Simonetti

**Affiliations:** 1Clinic of Dermatology, Department of Clinical and Molecular Sciences, Polytechnic University of Marche, 60126 Ancona, Italy; 2Clinic of Infectious Diseases, Department of Biomedical Sciences and Public Health, Polytechnic University of Marche, 60126 Ancona, Italy

**Keywords:** anti-microbial peptides, antibiotic resistance, Gram-positive Gram-negative, fungi

## Abstract

Antibiotic resistance is rapidly increasing, and new anti-infective therapies are urgently needed. In this regard, antimicrobial peptides (AMPs) may represent potential candidates for the treatment of infections caused by multiresistant microorganisms. In this narrative review, we reported the experience of our research group over 20 years. We described the AMPs we evaluated against Gram-positive, Gram-negative, and fungi. In conclusion, our experience shows that AMPs can be a key option for treating multiresistant infections and overcoming resistance mechanisms. The combination of AMPs allows antibiotics and antifungals that are no longer effective to exploit the synergistic effect by restoring their efficacy. A current limitation includes poor data on human patients, the cost of some AMPs, and their safety, which is why studies on humans are needed as soon as possible.

## 1. Introduction

Antibiotic resistance (AR) has for some decades been rapidly increasing and represents, nowadays, one of the world’s greatest challenges affecting animal and human health. The emergence of AR can be attributed not only to inappropriate antimicrobial prescriptions in humans but also to the overuse of antibiotics in animal breeding and agriculture [1,2]. 

Therefore, the identification of new anti-infective therapies is urgently needed. Over the past years, numerous efforts have been made to identify new molecules or new methods that could overcome the growing microbial resistance. In this perspective, antimicrobial peptides (AMPs) may represent potential candidates for the treatment of infections caused by multiresistant microorganisms. Inspired by this research area, our group at the “Polytechnic University of Marche” decided to evaluate the effectiveness of selected new AMPs over the last 20 years.

AMPs are oligopeptides usually composed of 12–50 cationic and hydrophobic amino acids with a positive net charge, representing essential components of innate immunity [3,4,5]. AMPs display broad-spectrum activity against a wide variety of pathogens, such as yeast, fungi, viruses, and bacteria [6,7].

In particular, many AMPs kill pathogens by interacting with negatively charged bacterial cell membranes; this leads to a change in their electrochemical potential, which generates cell membrane damage. Although several AMPs may also kill pathogens indirectly by modulating host immune responses [8,9,10,11], AMPs can also act with other different mechanisms of action, such as inhibiting communication between pathogens. These aspects will be described in the specific paragraphs that follow. Additionally, some AMPs show synergistic interactions with conventional molecules, contributing to the decrease in the selection of antibiotic-resistant bacteria and allowing us to restore sensitivity of conventional treatments [12,13]. The aim of this article is to summarise the 20 year experience of our research. Initially, our team included researchers from the Polytechnic University of Marche affiliated with the Clinic of Infectious Diseases, the Clinic of Surgery, the INRCA Experimental Animal Models for Aging Unit, and the Department of Inorganic Chemistry at the Medical University of Gdansk (Poland). Our scientific production was initially focused on the antimicrobial properties of AMPs in medical device infection, biofilm, and sepsis. With the arrival of researchers from dermatology and the Institute of Pathological Anatomy and Histology, our interest shifted to bacterial and fungal skin diseases. Our main goal was to provide new treatments to overcome multiresistant infections, particularly those caused by methicillin-resistant *Staphylococcus aureus* (MRSA) in chronic and surgical skin wounds, including burns. We studied the effects of AMPs not only for treating resistant bacteria-related infections in the skin but also for wound healing.

Simultaneously, we also continued to evaluate the potential and spectrum of action of selected AMPs against fungi and Gram-positive and Gram-negative bacteria. In the beginning, the experimental model was in vitro on colonies of resistant microorganisms taken from patients and then switched, in the case of effective AMPs, to the in vivo animal model. As our research progressed, we decided to assess the potential impact on wound healing by considering histological features and immunohistochemical markers that could quantify the action of AMPs vs. conventional antibiotics, such as vascular endothelium growth factor (VEGF), matrix metalloproteinases (MMP) expression, growth factors, or their receptors.

In the following paragraphs, we summarised the main areas of research performed, indicating the methods used and the results obtained.

## 2. AMPs and Biofilm in Medical Devices

Biofilm is characterised by bacterial cells that adhere to a substratum, interface, or each other and are embedded in a matrix of substances produced by the cells themselves. Biofilm often tends to develop on medical devices, in particular long-term silicone catheters such as the central venous catheter (CVC). The biofilm protects bacteria from antimicrobial therapy, leading to frequent failure of conventional antibiotic therapy [14,15,16].

Due to this issue, in the last few years different novel drug technologies have been studied, including antimicrobial peptides.

### 2.1. BMAP-28

BMAP-28 is a 27 residue peptide. It has an amidated C-terminus, and it has been shown to have the ability to kill bacteria in vitro. Furthermore, in vivo studies have also demonstrated BMAP-28 efficacy in reducing mortality in different infections [17,18].

Cirioni et al. (2005) assessed the efficacy of BMAP-28 pre-coating in the treatment of *S. aureus* central venous catheter-associated infections using the antibiotic-lock technique. In vitro studies revealed that pretreatment with BMAP-28 and then the use of antibiotics reduced the Minimal Bactericidal Concentration (MBC) values against biofilm. In vivo studies demonstrated that catheters pre-treated with BMAP-28 or high-dose antibiotics have a lower bacterial load compared to catheters with standard-dose antibiotics or without BMAP-28 (from 10^7^ to 10^3^ CFU/mL and bacteremia from 10^3^ to 10^1^ CFU/mL). A further significant reduction in bacterial load, from 10^7^ to 10^1^ CFU/mL, was observed when catheters were impregnated with BMAP-28 and then treated with a higher dose of antibiotics [19].

Another experimental study was performed to evaluate the efficacy of BMAP-28. In particular, the efficacy of BMAP-28 alone and in combination with vancomycin was assessed in animal models of ureteral stent infection induced by *Enterococcus faecalis* and *Staphylococcus aureus*. In vivo studies revealed that BMAP-28 reduced bacterial load (from 8 × 10^6^ to 5 × 10^4^ against *S. aureus* and from 8.7 × 10^6^ to 6.4 × 10^4^ against *E. faecalis*) and enhanced the effect of vancomycin (no bacterial count). This result suggests that the BMAP-28-impregnated ureteral stent has lower rates of infection. In vitro studies support these results [20].

### 2.2. Citropin 1.1

Citropin 1.1 is a wide-spectrum amphibian antimicrobial cationic peptide produced by the glands of the green tree frog, *Litoria citropa* [21].

An experimental study conducted in 2006 evaluated the efficacy of citropin 1.1, minocycline, and rifampicin in the prevention of *S. aureus* central venous catheter (CVC)-associated infection using the antibiotic-lock technique. In vitro studies show that biofilms were strongly affected by the presence of citropin 1.1, which also acts synergistically with hydrophobic antibiotics. In vivo studies confirm the same results; in fact, citropin 1.1 alone not only reduced bacterial load on the CVC from 10^7^ to 10^3^ CFU/mL but also enhanced the effect of commonly used antibiotics, reducing bacterial load to 10^1^ CFU/mL [22].

### 2.3. Temporin A

Temporin A is a hydrophobic, basic, antimicrobial peptide amide with antibiotic activity against a wide spectrum of microorganisms, including antibiotic-resistant Gram-positive cocci [23]. Temporin A is thought to act in conjunction with the formation of the ion channel in the bacterial cytoplasmic membrane [24].

Ghiselli et al. investigated the efficacy of temporin A as a prophylactic agent against methicillin sodium-susceptible (MS) and methicillin sodium-resistant (MR) *Staphylococcus epidermidis* vascular graft infection. In vitro studies revealed that both MR and MS were similarly susceptible to temporin A. In vivo studies support these results, showing that the use of a temporin A-soaked Dacron graft in vascular surgery can result in substantial bacterial growth inhibition (from 1.9 × 10^7^ to 3.4 × 10^3^ CFU/mL against *S. epidermidis* MS and from 3.9 × 10^7^ to 6.1 × 10^3^ CFU/mL against MR). Most of the antibiotic prophylactic treatments were helpful; nevertheless, only the association of a temporin A-soaked graft and intraperitoneal vancomycin hydrochloride inhibited bacterial growth for both the MR and MS strains [23].

Another study tested the efficacy of topical temporin A and RNAIII-inhibiting peptide (RIP) compared to rifampicin in preventing *S. epidermidis* and *S. aureus* graft infection in a rat pouch model [25].

RIP is a heptapeptide that has strong activity against *S. aureus* and *S. epidermidis*. Considering its mechanism of action, RIP inhibits cell-cell communication, also known as quorum sensing, preventing bacterial adhesion and virulence [26].

The results of this study showed that the use of temporin A-soaked and RIP-soaked Dacron grafts induced a significant bacterial growth inhibition. In fact, the combination of RIP and temporin A showed the lowest bacterial growth (negative quantitative cultures for VISE4 and from 6 × 10^7^ to 6.9 × 10^1^ CFU/mL for VISA4). More specifically, temporin A had a high antistaphylococcal activity, independent of the level of resistance shown by the isolates. RIP was more effective against staphylococcal strains when used alone than temporin A or rifampicin alone [25]. 

Giacometti et al. (2004) investigated the efficacy of temporin A soaking in combination with intraperitoneal linezolid in the prevention of vascular graft infection in a rat model of infection with *Staphylococcus epidermidis* with intermediate resistance to glycopeptides (GISE). The in vitro results show that temporin A and linezolid both have high activity against the GISE clinical strain. The in vivo study confirmed the strong activity against *S. epidermidis* of temporin A and linezolid, and it showed that the combination of temporin A with a parenteral antibiotic, such as linezolid (from 6.9 × 10^6^ to 3.8 × 10^2^ CFU/mL with linezolid and to 3.4 × 10^3^ with temporin A), may become a valid opportunity for chemoprophylaxis in vascular surgery [27].

Temporin A, citropin 1.1, CA (1-7)M (2-9)NH2, and Pal-KGK-NH2 were also studied in 2019 for their synergistic activity against methicillin-resistant *Staphylococcus aureus* (MRSA) biofilms developed on polystyrene surfaces (PSS) and central venous catheters (CVC). The study highlighted that antimicrobial peptides have strong activity in inhibiting biofilm formation on both PSS (citropin 1.1 inhibited biofilm formation of all MRSA strains tested) and CVC (citropin 1.1 caused a biomass reduction for the reference strain with an OD_570_ of 0.152 compared with the control). The eradication of preformed biofilms, on the other hand, was more difficult and took 24 hours after contact between the AMP and biofilms. The combination of AMP had synergistic activity, leading to an improvement in the performance of all the peptides in the removal of biofilms [28].

### 2.4. Other Peptides

Polycationic peptides have been studied in recent years for their antimicrobial activity. Buforin II and ranalexin are polycationic peptides derived from amphibian tissues. Cerioni et al. investigated the efficacy of ranalexin and buforin II in the prevention of vascular prosthetic graft infection due to *Staphylococcus epidermidis* with intermediate glycopeptide resistance.

Both peptides demonstrate strong in vitro activity. In vivo studies demonstrated that buforin II and ranalexin (from 4.9 × 10^6^ to 1.9 × 10^2^ CFU/mL) had a stronger activity than vancomycin (from 4.9 × 10^6^ to 6.2 × 10^3^ CFU/mL) and teicoplanin (from 4.9 × 10^6^ to 5.1 × 10^4^ CFU/mL). In particular, buforin II was able to inhibit bacterial growth completely [29].

Another study compared the activity of the same polycationic peptides to that of rifampicin in the prevention of methicillin-susceptible and methicillin-resistant *Staphylococcus epidermidis* vascular prosthetic graft infections. This study found that polycationic activities against *Staphylococcus epidermidis* were comparable to rifampicin. The combinations of buforin II and ranalexin-coated grafts with cefazolin showed stronger activity against the methicillin-resistant strain (no evidence of infection) than that of the combination of rifampicin-coated grafts and cefazolin (10^2^ bacterial growth) [30].

Pal-Lys-NH_2_ and Pal-Lys-Lys are lipopeptides with bactericidal activity, and they are effective against different Gram-positive cocci [31].

A study investigated their action alone or in combination with vancomycin in the prevention of prosthesis biofilm in a subcutaneous rat pouch model of staphylococcal vascular graft infection. The results of this study showed that vancomycin (from 6.94 log to 3.65 log CFU/mL) and lipopeptides (from 6.94 log to 3.87 log CFU/mL for Pal-Lys-Lys NH_2_ and from 6.94 log to 4.080 log CFU/mL for Pal-Lys-Lys) when used alone had similar activity. The combination of vancomycin with Pal-Lys-Lys-NH_2_ had the strongest efficacy (from 6.94 log to 1 log CFU/mL). The in vitro study globally confirms the in vivo one [32].

Distinctin is an antimicrobial peptide with a heterodimeric structure. It has a lytic activity on unilamellar vesicles, suggesting their possible action on bacterial membranes [33].

In a study, the efficacy of distinctin was assessed in the treatment of *Staphylococcus aureus* CVC-associated infection, in particular in inhibiting the attachment of *S. aureus* to CVCs and in increasing its susceptibility to glycopeptides and carbapenems once it is adherent. The in vitro study showed a valid activity of distinctin on the biofilm and the ability to enhance the efficacy of antibiotics when used in combination. In vivo studies confirmed these results; furthermore, treatment with antibiotics and distinctin induced a significant reduction in bacterial loads on the CVC (from 10^6^ to 10^1^ CFU/mL) with no evidence of bacteriemia [34].

Protegrins are cysteine-rich AMPs and comprise 16–18 amino acids. IB-367 is a synthetic protegrin with bactericidal and fungicidal activity [35].

Ghiselli et al. evaluated the efficacy of a pre-treatment with IB-367 and its capacity for enhancing the efficacy of linezolid on Gram-positive biofilm in animal models of CVC infection. The study showed that IB-367 pre-treatment of CVC enhanced linezolid activity, causing a higher biofilm bacterial load reduction (from 10^6^ to 10^1^ CFU/mL) and absence of bacteriemia. In conclusion, IB-367 could be considered an interesting adjunctive agent to conventional antibiotics for the treatment of CVC and other medical devices [36].

Cirioni et al. investigated the efficacy of daptomycin and rifampicin alone and in combination in the prevention of vascular graft biofilm formation in a rat pouch model of Staphylococcal infection. Rifampicin is a glycopeptide antibiotic, while daptomycin is a lipopeptide. The results of this study showed that both rifampicin and daptomycin have good efficacy when used alone (from 7.4 × 10^6^ to 3.3 × 10^2^ CFU/mL for daptomycin and from 7.4 × 10^6^ to 4.2 × 10^3^ CFU/mL for rifampicin). When they are used in combination, their efficacy is higher than that of each single compound (from 7.4 × 10^6^ to 10^1^ CFU/mL). Moreover, their combination prevented the emergence of rifampicin resistance in adherent bacteria. These results were confirmed by in vitro studies [37].

Another study investigated the efficacy of levofloxacin, cefazolin, and teicoplanin in preventing vascular prosthetic graft infection induced by methicillin-susceptible and methicillin-resistant *Staphylococcus epidermidis*. The three compounds had similar efficacies, but levofloxacin (from 10^6^ to 10^3^ CFU/mL) showed slightly less efficacy than teicoplanin (from 10^6^ to 10^2^ CFU/mL) against the methicillin-resistant strain. Furthermore, the results highlighted that the most useful combination for the prevention of late-appearing vascular graft infections is rifampicin-levofloxacin (no infection detected). However, rifampicin-teicoplanin was also very effective (no infection was detected) [38].AMPs, in our experience, have a high efficacy in reducing bacterial load on the surface of medical devices; this efficacy is frequently comparable to that of the most commonly used antibiotics. Furthermore, when AMPs are used in combination with other antibiotics, they increase their efficacy, leading to no evidence of bacteriemia in most cases.

## 3. AMPs and Gram-Positive Sepsis

Sepsis represents a serious clinical problem given its severity, prevalence, and difficulty in treatment. Specifically, in 50% of cases, sepsis results from Gram-positive infections. The most frequently involved microorganisms are *Staphylococcus aureus* and *Streptococcus epidermidis*. Antibiotic therapy is not always effective, partly because of the increasing prevalence of antibiotic resistance. For this reason, new molecules such as antimicrobial peptides are increasingly being considered [39].

### 3.1. Distinctin

Distinctin is an amphipathic antimicrobial peptide that has a structure characterised by two different peptide chains connected by a disulfide bridge. It has been isolated from the skin of *Phyllomedusa distincta* and has shown good antimicrobial activity in vitro. The in vitro efficacy was also confirmed in vivo. In fact, this molecule demonstrated efficacy when administered intravenously in neutropenic mouse models infected with *Staphylococcus aureus*, either alone or in combination with other antibiotics. Notably, its efficacy was shown to be highest when administered together with glycopeptides in the absence of toxic events related to the administration of the peptide itself. Distinctin in combination with vancomycin and teicoplanin resulted in the lowest lethality rate in the aforementioned models [39].

### 3.2. Temporin A

Temporin A has demonstrated the ability to inhibit the production of TNF-alpha, IL-6, and NO by macrophages in mouse models and is active against antibiotic-resistant Gram-positive cocci. Specifically, it has shown efficacy against *Staphylococcus aureus* in mouse models and was particularly high 6 h after injection. The most effective antibiotic used in combination was imipenem (lethality rates of 25% for temporin A, 20% for imipenem, and 10% for temporin A + imipenem). Temporin A is able to facilitate the passage of imipenem across the bacterial membrane by destructuring it, as both act on peptidoglycan. In addition, temporin A appears to induce the migration of human monocytes, neutrophils, and macrophages [40].

### 3.3. BMAP-28

BMAP-28 has been shown to inhibit TNF-alpha release and NO production. In mouse models, it has shown similar lethality to antibiotics such as clarithromycin and imipenem against *Staphylococcus aureus*. In addition, it appears to have a superior ability to neutralise bacterial products released by Gram-positive bacteria, a positive factor in severe staphylococcal infections when used in combination with other antimicrobial agents [41].

### 3.4. Magainin II and Cecropin A

Magainin II and Cecropin A are two alpha-helical antimicrobial peptides that have demonstrated in vitro activity and in vivo efficacy against *Staphylococcus aureus* with intermediate resistance to glycopeptides in combination with vancomycin. In particular, the combination of magainin II and vancomycin has been shown to be particularly effective in reducing lethality in murine models of staphylococcal sepsis (lethality of 1/20 vs. 6/20 vancomycin vs. 10/20 magainin II vs. 12/20 cecropin A). These two peptides appear to be able to insert into the cytoplasmic membrane and activate murine bacterial hydrolases, resulting in peptidoglycan damage and cell lysis [42].

### 3.5. Aurein 1.2

This is an amphipathic, alpha-helical peptide composed of only 13 amino acids. It has demonstrated antimicrobial activity in vitro against Gram-positive cocci at concentrations ranging from 1 to 16 mg/litre and synergistic activity when administered in combination with clarithromycin and minocycline. In particular, aurein 1.2 exerts its action by making the membrane more permeable and less organised, allowing the entry of hydrophobic substrates [43].

Other peptides that demonstrated antimicrobial activity in vitro against Gram positive cocci include palmitol (pal)-lys-lys-NH_2_ and pal-lys-lys, uperine 3.6, and citropin 1.1.

The lipopeptides showed in vitro efficacy mainly against enterococci and streptococci compared with Staphylococci and *Rhodococcus equi*. Their efficacy was higher when combined with beta-lactams and vancomycin; they also proved effective against both susceptible and multidrug-resistant germs.

Uperine 3.6 is a broad-spectrum antibiotic peptide isolated from the Australian amphibian *Uperoleia mjobergii*. It consists of only 17 amino acids, and for this reason, represents the smallest of the antibiotic peptides isolated from amphibians. Although most of the antibiotics tested were more effective than uperine 3.6, it was effective against both susceptible and multiresistant germs [44].

Citropin 1.1 is an antimicrobial peptide derived from the Australian frog *Litoria citropa*. It is a small peptide consisting of 16 amino acids produced by both the dorsal and submental glands of *Litoria citropa*. It has been shown to be effective in vitro against 12 nosocomial isolates of *Rhodococcus equi* at concentrations ranging from 2 to 8 mg/L. The peptide presented synergistic activity against this germ when combined with clarithromycin, doxycycline, and rifampicin. In addition, other in vitro studies have demonstrated its efficacy against staphylococci and streptococci at concentrations ranging from 1 to 16 mg/L. Synergy was demonstrated when citropin 1.1 was combined with hydrophobic antibiotics such as clarithromycin and doxyxline [45,46].

## 4. AMPs and Wound Infection

We report on four studies whose purpose was to evaluate the efficacy of new experimental AMPs in the management of infected surgical wounds in mouse models. The infection was established using MRSA, the most frequent aetiological agent in cSSTIs. In these studies, the role of antimicrobial peptides in infected wounds was evaluated mostly by considering bacterial growth (quantitative cultures of excised tissues) and healing parameters such as collagen organisation, degree of re-epithelialisation, granulation tissue formation, angiogenesis, and VEGF expression. The results were compared with data from control groups, such as animals with no infected wounds (treated and untreated), animals with infected but untreated wounds, and animals treated with other conventional antibiotics.

In 2013, Cirioni et al. studied the activity of the AMP IB-367 (a synthetic protegrin) as an immunomodulator and immune enhancer, evaluating whether pretreatment with this peptide resulted in an enhancement of antibiotic efficacy (daptomycin and teicoplanin) against MRSA wound infection in a mouse model. The main outcome measures were quantitative bacterial culture and analysis of natural killer (NK) cytotoxicity and leukocyte phenotype. Antibiotics alone showed a comparable antimicrobial efficacy, but their association with IB-367 significantly enhanced the antimicrobial activities. When compared to antibiotics alone (2 log reduction), IB-367 plus daptomycin showed a 4 log reduction in bacterial growth, with IB-367 plus daptomycin showing the highest efficacy (reduction in bacterial load of 2.7 × 10^3^ ± 0.3 × 10^3^ c.f.u. mL^-1^). IB-367 action is also related to a modulation of the innate immune response, mainly represented by an increase in NK cell activity (but not NK cell number) and increasing the number of both CD11b and Gr-1 cells 3 days after MRSA challenge, over the levels observed in the respective controls [47].

LL-37 is the only human antimicrobial peptide that belongs to the cathelicidins. LL-37 showed a broad-spectrum against several different pathogens, such as Gram-positive and Gram-negative bacteria, viruses, and fungi [48]. Moreover, LL-37 revealed other biological activities, such as regulation of responses to inflammation and demonstration of an important activity in wound closure and angiogenesis [49].

Simonetti et al. (2021) evaluated the efficacy of synthetic cathelicidin LL-37 in MRSA-infected surgical wounds in mice, in comparison with teicoplanin treatment. The results of the study showed that LL-37 had a stronger effect than teicoplanin on the wound healing process in MRSA-infected mice, although with a lower effect on bacterial culture growth. LL-37 reduced bacterial numbers to 7.1 × 10^5^ ± 0.6 × 10^5^ CFU/g, and 6.9 × 10^2^ ± 0.4 × 10^2^ CFU/g when combined with topical LL-37, while i.p. teicoplanin produced a bacterial count of 7.4 × 10^4^ ± 1.0 × 10^4^ CFU/g and 3.0 × 10^2^ ± 1.2 × 10^2^ CFU/g when associated with topical teicoplanin. LL-37, after topical and parenteral administration, enhanced wound closure via stimulation of granulation tissue formation associated with a better angiogenesis process and better organised collagen deposition and reconstitution of the epithelium in comparison with the teicoplanin treatment group [50].

Another study evaluated the efficacy of distinctin, a heterodimeric peptide from the Amazonian frog *Phyllomedusa distincta*, in the management of cutaneous MRSA wound infections in an experimental mouse model. It was found that topical distinctin combined with parenteral teicoplanin inhibited bacterial growth to 3.4 × 10^1^ ± 1.0 × 10^1^ (levels comparable with those observed in uninfected animals), but the combination of topical and parenteral teicoplanin proved to be the most effective in reducing bacterial counts (4.7 × 10^1^ ± 1.6 × 10^1^). Furthermore, when compared to mice treated only with teicoplanin, wounded areas of animals treated with distinctin were characterised by a more mature granulation tissue, a more organised and denser type of connective tissue, and a significant reduction in fibrinous exudation [51].

In 2007, Simonetti et al. conducted a study on temporin A, investigating the effect of its topical use in mouse models of MRSA-infected wounds. Temporin-A treatment combined with teicoplanin injection significantly reduced the bacterial load to 0.85 × 10^1^ ± 0.1 × 10^1^ CFU/mL. Histological examination showed that infected mice receiving temporin A-soaked Allevyn (with or without teicoplanin) had a higher degree of granulation tissue formation and collagen deposition compared to the other treated groups. A significant increase in serum VEGF expression was observed in mice receiving temporin A topically with or without intraperitoneal teicoplanin [52].

### Wound Infection: Commercially Available AMPs and Perspectives

In this section, we mention studies that analyse the management of MRSA wound infections in mouse models with commercially available antimicrobial peptides such as teicoplanin, daptomycin, and dalbavancin.

Ghiselli et al. (2006) wanted to compare the efficacy of topical vs. systemic teicoplanin for the treatment of wound infection with *Staphylococcus aureus* in a mouse model. Data analysis showed that strong inhibition of bacterial growth was achieved in all groups treated with intraperitoneal teicoplanin. However, the highest inhibition of bacterial growth was obtained in the group that received teicoplanin-soaked Allevyn and intraperitoneal teicoplanin. Histological examination showed that each treatment modality was able to reduce the delay in wound repair, but the best results were obtained with teicoplanin-soaked Allevyn, with and without intraperitoneal teicoplanin, associated with a wound remodelling similar to that of not-infected mice (assessing microvessel density, VEGF expression, and granulation tissue formation in tissue sections) [53].

Daptomycin is a bactericidal lipopeptide antibiotic widely used to treat systemic infections caused by Gram-positive cocci [54]. In a study conducted by Simonetti et al. (2017), a mouse model was used to study the in vivo efficacy of daptomycin in the treatment of burn wound infections caused by *S. aureus* and evaluate the wound healing process through morphological and immunohistochemical analysis, compared to teicoplanin. The highest inhibition of infection in terms of bacterial load was achieved in the group that received daptomycin (2.0 × 10^3^ ± 0.4 × 10^3^ CFU/g), which also showed better overall healing with epithelialisation and significantly higher collagen scores than the other groups, and these findings were also confirmed by immunohistochemical data on EGFR and FGF-2. The results of this in vivo study in an animal model showed that daptomycin demonstrated stronger antimicrobial activity than teicoplanin [55]. Moreover, daptomycin, in a previous study, showed synergy in its effect against MRSA when combined with other antibiotics such as tigecycline [56].

Dalbavancin is a semisynthetic lipoglycopeptide active against Gram-positive bacteria and has been approved for the treatment of acute bacterial skin and skin structure infections (ABSSSI) [57]. In a 2020 study conducted by Simonetti O. et al., the aim was to determine the effect of dalbavancin administration on wound healing compared to that of vancomycin and to elucidate if and how EGFR, MMP-1, MMP-9, and VEGF could be involved in its therapeutic mechanisms. A mouse model of MRSA skin infection was established, and mice were treated daily with vancomycin or weekly with dalbavancin at days 1 and 8. Both dalbavancin and vancomycin were effective in reducing the bacterial load (8.71 × 10^5^ ± 9.02 × 10^5^ CFU/mL vs. 8.04 × 10^6^ ± 7.96 × 10^6^ CFU/mL, respectively). The wounds treated with dalbavancin showed well-organised granulation tissue with numerous blood vessels, although slightly less than that in the uninfected group. The immunohistochemical staining also showed elevated EGFR and VEGF expression in both treated groups (higher in dalbavancin-treated mice), decreased MMP-1 and MMP-9 levels in uninfected tissue, and in both treated tissues compared with untreated infected wounds. This study revealed faster healing with dalbavancin treatment than might be associated with higher EGFR and VEGF levels, with the lowest values of MMP-9 and MMP-1 expression [58].

## 5. AMPs and *Enterococcus faecalis* Infection

Enterococci are responsible for multiple nosocomial infections, and they have a high frequency of multidrug infections. The majority of enterococcal infections are caused by *Enterococcus faecalis,* which is often resistant to multiple antibiotics. Thus, it is very important to search for new antimicrobial compounds such as AMPs [59].

Giacometti et al. (2004) evaluated the in vitro activity of temporin A against *E. faecalis* nosocomial isolates, including vancomycin-resistant strains, and its in vitro activity combined with antibiotics. Temporin A demonstrated potent antibacterial activity against *E. faecalis*, including vancomycin-resistant strains, in vitro, especially when combined with amoxicillin/clavulanic acid and imipenem. In conclusion, this peptide could be used in the future as an adjuvant in the therapy for enterococcal infections [60].

Cirioni et al. conducted an experimental study to evaluate both the in vitro and in vivo interaction between the Laur-CKK-NH_2_ lipopeptide and daptomycin using two *Enterococcus faecalis* strains with different patterns of susceptibilities. The in vitro experiments showed that the Laur-CKK-NH_2_ dimer is able to prevent the emergence of daptomycin resistance. Additionally, for in vivo studies using a mouse model of enterococcal sepsis, the Laur-CKK-NH_2_ dimer and daptomycin exhibited the highest efficacy in measuring lethality and bacteremia [61].

## 6. AMPs and Gram-Negative Bacteria

Infections sustained by multi-drug-resistant (MDR) Gram-negative bacteria represent one of the most important therapeutic challenges, considering that their resistance to antibiotics is expanding from extended-spectrum beta-lactamases and carbapenemases [62] to the *mcr* gene, which is responsible for colistin resistance [63]. This is why new molecules need to be evaluated in order to overcome AMRs. AMPs can also be a valuable aid in the treatment of Gram-negative infections.

### 6.1. Protegrin-1

*Acinetobacter baumannii*, in our experience, is a Gram-negative pathogen with a high risk of developing multiple antibiotic resistances, particularly in the hospital setting and in immunocompromised patients. Although it has been shown that the combination of colistin with daptomycin or teicoplanin can make antibacterial therapy effective in a mouse model [64], colistin may not be sufficient in cases of *A. baumannii* MDR. In an in-vitro model of cultures of *A. baumannii* MDR, also resistant to colistin, taken from surgical wounds, the minimum inhibitory concentration (MIC), 2 mcg/mL, and minimum bactericidal concentration (MBC), 8 mcg/mL, of Protegrin-1 (PG-1) were assessed. PG-1 is an 18-amino-acid beta-hairpin AMP belonging to the cathelicidin family, with excellent bactericidal action in monotherapy and excellent synergy with colistin. No resistance to PG-1 developed, but there was also no effect on biofilm. However, PG-1 is proposed as an interesting future perspective in gram-negative MDR infections [65].

### 6.2. Pexiganan

Another interesting AMP is pexiganan, a 22-amino acid synthetic magainin-based lysine-rich peptide that showed effective action against *A. baumannii* in a mouse model of sepsis. Both the groups treated with pexiganan (1 mg/kg intraperitoneal) and colistin (1 mg/kg intraperitoneal) showed good antibacterial efficacy, but the lowest bacterial count occurred in the pexiganan plus colistin combination group, which also achieved the highest survival rate (90%) [66]. This AMP may also help overcome MDR phenomena involving last-line antibiotics such as colistin.

In addition, pexiganan was shown to be synergistic with tigecycline in a mouse model infected with *Pseudomonas aeruginosa*, making a normally ineffective antibiotic effective against Gram-negative bacteria [67]. This offers new perspectives, considering the possible use of antibiotics that would not normally be effective against Gram-negative bacteria. In another study on a mouse model with urethral stents, the effect of pexiganan and imipenem at sub-MIC concentrations on the biofilm produced by *P. aeruginosa*, a slime producer was evaluated [68]. Studying biofilm production in vitro, the group of mice treated with pexiganan and imipenem showed a marked reduction in adhesion and biofilm expression compared to untreated controls (average reductions of 34 ± 8% and 27 ± 4%, respectively), highlighting a role for this AMP in the management of infections sustained by *P. aeruginosa* capable of producing biofilm.

### 6.3. Alpha-Helical AMPs

Magainin II and cecropin A, alpha-helical AMPs, were used in vitro and in vivo in a mouse model against *P. aeruginosa* MDR [69]. Magainin II and cecropin A showed a strong antimicrobial action, achieving a significant reduction in plasma endotoxin (≤0.015 ± 0.0 EU/mL and ≤0.015 ± 0.0 EU/mL, respectively) and TNF-alpha concentrations (0.38 ± 0.02 ng/mL and 0.44 ± 0.03 ng/mL, respectively) compared to control (38.40 ± 2.89 EU/mL and 145.16 ± 18.32 ng/mL) and rifampicin-treated groups (29.45 ± 3.43 EU/mL and 98.0 ± 8.7 ng/mL). The latter, in monotherapy, showed no action against *P. aeruginosa*, as in other studies [70] while the combination of magainin II and cecropin A was proved significantly effective in reducing bacterial counts and mortality. This study highlights how the combination of AMPs and antibiotics that are normally ineffective against *P. aeruginosa* may be a novel solution for new therapeutic needs.

### 6.4. Tachyplesin III

Tachyplesin III, a potent disulphide-linked peptide, showed synergistic action in vitro with beta-lactams and colistin against *P. aeruginosa* MDR [71]. In a study by Cirioni et al. [72], the activity and in vivo efficacy of Tachyplesin III, colistin, and imipenem against a multiresistant *P. aeruginosa* strain, were investigated in a murine model of sepsis. Bacteremia levels were significantly lower in the combination therapy groups (1.1 × 10^1^ ± 0.1 × 10^1^ CFU/mL, Tachyplesin III and imipenem), (4.6 × 10^1^ ± 0.5 × 10^1^ CFU/mL colistin and imipenem) than in the single agent groups (control 5.8 × 10^7^ ± 0.8 × 10^7^ CFU/mL, Tachyplesin III 3.6 × 10^3^ ± 0.6 × 10^3^ CFU/mL), in particular Tachyplesin III with imipenem had the highest efficacy in terms of bacterial lethality, quantitative blood cultures, and plasma levels of lipopolysaccharide, tumour necrosis factor alpha, and interleukin-6. Once again, combination therapy with AMPs and traditional antibiotics proved to be a very useful option. Additionally, in a study with piperacillin/tazobactam (TZP), the authors [73] reported how mice treated with Tachyplesin III in combination with TZP demonstrated the greatest efficacy compared to monotherapy, implying that a urethral stent coated with Tachyplesin III can reduce *P. aeruginosa* bacterial growth by 1,000-fold.

Finally, the effects of Tachyplesin III and clarithromycin were studied in a mouse model of *Escherichia coli* sepsis. It was seen that Tachyplesin III (1 mg/kg intraperitoneally) alone resulted in greater antimicrobial action and a significant reduction in endotoxin and TNF-alpha plasma concentrations compared to the control and clarithromycin (50 mg/kg intraperitoneally) alone groups. The latter showed no antimicrobial activity but resulted in the reduction of endotoxins and TNF-alpha plasma concentrations. The combination group of Tachyplesin III and clarithromycin was seen to be the most effective in all parameters analysed [74].

### 6.5. LL-37 and Tritrpticin

LL-37, a human cathelicidin, showed an interesting anti-pseudomonas action in neutropenic patients. In a neutropenic mouse model, septic shock was induced by *P. aeruginosa*, and then the groups were randomised into those treated with placebo, imipenem, granulocyte CSF (G-CSF), LL-37 + G-CSF, or imipenem + G-CSF. Although all therapy groups were superior to the control, the LL-37 + G-CSF group was the most effective in preventing sepsis by significantly lowering neutrophil apoptosis in vitro. The authors [[75], Figure 2] reported similar results were obtained by *Escherichia coli* 0111:B4 LPS and ATCC 25922 in the murine animal model of sepsis. The authors [76] used treatment groups which consisted of LL-37, polymyxin B, imipenem, or piperacillin vs. placebo. Despite the fact that all treatments reduced lethality, only LL-37 and polymyxin B showed a reduction in endotoxin (≤0.015 ± 0.0 EU/mL and ≤0.015 ± 0.0 EU/mL, respectively, vs. 33.49 ± 4.69 ng/mL piperacillin) and TNF-α plasma levels (0.22 ± 0.01 ng/mL and 0.21 ± 0.01 ng/mL, respectively, vs. 131.12 ± 17.0 ng/mL piperacillin, see Figure 2 from [77]). Furthermore, there were no statistically significant differences in antimicrobial and antiendotoxin activities between LL-37 and polymyxyn B. Given its anti-inflammatory effect, LL-37 is an excellent candidate for the treatment of Gram-negative sepsis. Furthermore, LL-37 was combined with colistin against multidrug-resistant *Escherichia coli*, demonstrating good activity in reducing biofilm formation [77].

Another cathelicidin that has shown activity in vitro against *P. aeruginosa* MDR is tritrpticin, which completely inhibits the procoagulant activity of lipopolysaccharides and shows a synergistic effect with beta lactams [78].

### 6.6. IB-367

The efficacy of topical IB-367, a 17-amino acid synthetic protegrin, was evaluated in a mouse model [79] of a skin wound infected with *P. aeruginosa* or *E. coli*, both MDR, alone and in combination with colistin or imipenem (intraperitoneal). The group treated with IB-367 plus colistin showed the greatest inhibition of both bacterial strains, demonstrating excellent efficacy. In vitro, IB-367 inhibited both bacterial strains with a rapid killing time of 40 min. Therefore, IB-367 may be an excellent candidate for topical therapy of Gram-negative infected wounds in the future.

### 6.7. Citropin 1.1

Citropin 1.1 is an amphibian peptide studied alone and in combination with tazobactam-piperacillin (TZP) in a mouse model of *E. coli* sepsis. When compared to controls, all treatment groups—intraperitoneal 1 mg/kg cytropin 1.1, 120 mg/kg TZP, or 1 mg/kg cytropin 1.1 plus 60 mg/kg TZP—reduced lethality. The group with cytropin 1.1 alone showed a significant reduction in plasma endotoxins and inflammatory cytokines, while TZP exerted the opposite effect. The combination of cytropin 1.1 and TZP was most effective in reducing lethality, bacterial growth in blood and peritoneum, and oxidative stress indices in plasma. Citropin 1.1 is therefore an AMP with not only antimicrobial but also immunomodulatory properties and may be an interesting option in conditions of severe Gram-negative infection [80].

All of these AMPs are molecules that showed in vivo action against Gram-negative MDR bacteria, suggesting a possible use to overcome increasing antibiotic resistance, as proposed by other in vitro studies. [79,80,81,82,83] However, only some of these molecules, in our opinion, can be developed for use in humans. Particularly in Gram-negative skin infections, an interesting role could be played by IB-367 as the only topical agent to be combined with traditional therapy. Further studies in patients are needed to evaluate both the efficacy and safety of these molecules in humans.

### 6.8. Colistin

Colistin, previously studied in combination with pexiganam against Gram-negative bacterial infection [67], was also combined with teicoplanin or daptomycin in an experimental mouse model of multiresistant *Acinetobacter baumannii* infection. The permeabilising effect of colistin on the *A. baumannii* outer membrane allows glycopeptide and lipopeptide molecules to enter, which are normally excluded due to their size, resulting in a better patient outcome in severe infections caused by multiresistant microorganisms like *A. baumannii* (6.7. × 10^4^ ± 1.1 × 10^4^ colistin alone, 5.0 × 10^9^ ± 1.6 × 10^9^ daptomycin alone, 7.3 × 10^9^ ± 1.8 × 10^9^ teicoplanin alone, 2.9 × 10^2^ ± 0.4 × 10^2^ colistin + daptomycin, and 3.1 × 10^2^ ± 0.2 × 10^2^ colistin + teicoplanin) [83].

## 7. AMPs and Fungi

Fungal infections represent one of the most frequent public health problems [84], also considering the progressive increase in resistance to traditional therapies and the side effects of some antimycotics that limit their use, especially in immunocompromised patients [85].

For this reason, it is necessary to evaluate new molecules to expand our therapeutic options. In our experience, we also evaluated the action of AMPs against fungal infections. Some AMPs show both antibacterial and antibiotic action and can therefore be excellent options in the treatment of mixed infections, allowing them to act on different mechanisms of action than current therapies, which mainly affect sterol biosynthesis [84,85,86,87,88,89].

### 7.1. IB-367

IB-367 is a protegrin with activity against Gram-negative bacteria as well as fungi. In an in vitro study, the efficacy of IB-367 alone and in combination with fluconazole, itraconazole, and terbinafine was evaluated against strains from patients infected with *Trichophyton mentagrophytes*, *T. rubrumand*, and *Microsporum canis*. In monotherapy, the lowest MIC was for terbinafine and itraconazole, but there was a synergy of 35% with IB-367/fluconazole, 30% with IB-367/ITRA, and 25% with IB-367/TERB [88]. This study suggests that IB-367 may be a molecule capable of increasing the efficacy of currently available antifungal therapies.

In addition, IB-367 showed in vitro a rapid fungicidal action against *Candida* spp., both sensitive and resistant to fluconazole. Synergistic action occurred in 41.6% of cases with fluconazole and 44% of cases with amphotericin B, without antagonism [89]. For these reasons, IB-367 is also a very promising molecule for treating candida infections.

### 7.2. Lipopeptide PAL-Lys-Lys-NH_2_

The short lipopeptide palmitoyl PAL-Lys-Lys-NH_2_ (PAL) was evaluated in vitro against *Candida* spp. alone and in combination with fluconazole, amphotericin B, and caspofungin. All drugs showed good activity against *Candida* strains; however amphotericin B and caspofungin had the lowest MIC. PAL showed relevant synergistic action with PAL/fluconazole (81.25%), PAL/amphotericin B (75%), and particularly with PAL/caspofungin (87.5%) [90]. In our opinion, the combination of PAL/caspofungin may be a new therapeutic option in cases of severe candida infection.

In cases of severe *Cryptococcus neoformans* infections, PAL was also effective in vitro, showing synergy in 21.4% of cases with amphotericin B, suggesting its possible use in infected patients to reduce the dosage and side effects of amphotericin B [91].

PAL was also studied in vitro against several clinical isolates of *dermatophytes* [92,93]. PAL and fluconazole showed a lower MIC and lower fungal biomass than gamma-terpinene, a component of tea tree oil. Finally, PAL was superior to fluconazole in reducing hyphal viability against both dermatophytes, suggesting its possible role in the treatment of these fungi as well.

### 7.3. Tachyplesin III

As seen against Gram-negatives, Tachyplesin III was also evaluated in mycotic dermatophyte infections in vitro. Terbinafine had a significantly lower MIC than Tachyplesin III (*p* < 0.001). The combination of the two therapies showed synergistic activity in 30% of cases, and no antagonism was recorded. Interestingly, both Tachyplesin III and terbinafine significantly reduced growth in *M. canis* (*p* < 0.01) [94]. This AMP could therefore be useful in combination with terbinafine to lower the dose of the antifungal while maintaining efficacy and safety.

### 7.4. C14-NleRR-NH_2_ and C14-WRR-NH_2_

Two lipopeptides (C14-NleRR-NH_2_ and C14-WRR-NH_2_) were studied to assess the antifungal activity against azole-resistant *Aspergillus fumigatus*. Both lipopeptides showed antifungal activity, with MICs ranging from 8 to 16 mg/L, and a dose-dependent effect was confirmed by both time-kill curves and XTT assays. Moreover, microscopy showed that hyphae growth was hampered at concentrations equal to or higher than MICs. Our results showed that both C14-NleRR-NH_2_ and C14-WRR-NH_2_ are effective against the resistant isolates tested, and this further prompts the research of lipopeptides as alternatives in antifungal therapy [95].

Finally, our experience also includes studies [96,97] evaluating the in vitro efficacy of AMPs on *Pneumocystis carinii* taken from patients with pneumonia. Cecropin P1, magainin II, indolicidin, and ranalexin alone and in combination with macrolides and dihydrofolate reductase inhibitors (DHFRs) were investigated. The four peptides suppressed the growth of *P. carinii* with no change in combination, with the exception of ranalexin, which showed synergistic activity with the macrolide [96]. Furthermore, an in vitro study on cell monolayers revealed that cecropin P1 and magainin II may be effective in inhibiting *P. carinii* growth at non-toxic cell monolayer concentrations [97].

## 8. Conclusions

In conclusion, our experience shows that AMPs can be a key option for treating infections sustained by multiresistant microorganisms and overcoming resistance mechanisms against currently used antibiotics or antifungals. The combination of AMPs allows antibiotics and antifungals that are no longer effective due to resistance to exploit the synergistic effect by restoring their efficacy. This will be crucial in the near future, considering the growing spread of antibiotic resistance. In conclusion, our experience shows that AMPs can be a key option for treating infections sustained by multiresistant microorganisms and overcoming resistance mechanisms against currently used antibiotics or antifungals. The combination of AMPs allows antibiotics and antifungals that are no longer effective due to resistance to exploit the synergistic effect by restoring their efficacy. Our research has shown that the peptides allow the penetration of antibiotic molecules inside the bacterial bodies, which would otherwise be primarily ineffective against those bacterial species, thus allowing an antibiotic action in some ways “unexpected” as in the case of some beta-lactams, macrolides, or tetracyclines when combined with peptides for the treatment of Gram-negative microorganism infections.

This will be crucial in the near future, considering the growing resistance.

Moreover, our studies and other more recent ones have highlighted the possibility of coating some devices with peptides (we did it by hand by immersing the device, e.g., Dacron, in a peptide solution), and now other researchers do it with covalent bonds. This possibility can be exploited, for example, for orthopedic prostheses, long-lasting catheters, etc. A current limitation is the lack of data on human patients, the high cost of some AMPs, and their safety (which is improving thanks to cytotoxicity studies on cell monolayers). In addition, the problem of the frequent peptides’ short half-life must be considered. This issue will have to be addressed in the future by seeking solutions similar to those obtained with glycopeptides such as dalbavancin and oritavancin, glycolipopeptides with prolonged half-lives (250–350 h), allowing once-weekly (dalbavancin) administration or a unique single-dose regimen (oritavancin). 

## Data Availability

Not applicable.

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
