# Peer review of "Our Experience over 20 Years: Antimicrobial Peptides against Gram Positives, Gram Negatives, and Fungi"

_pharmaceutics, 2022, doi:10.3390/pharmaceutics15010040_

Round 1

Reviewer 1 Report

*The introduction does not tell the audience the problem statement's history and the tendency in this field.

* The recently published paper deal with  MOFs and Heterocyclic compounds as  a new antimicrobial classes  not involved and must be cited:

https://link.springer.com/article/10.1007/s13738-022-02595-8

https://www.sciencedirect.com/science/article/abs/pii/S0022286021018536

https://www.sciencedirect.com/science/article/abs/pii/S101060302100441X

https://link.springer.com/article/10.1007/s10570-021-04147-4

https://www.sciencedirect.com/science/article/pii/S2405844020317990

https://www.sciencedirect.com/science/article/abs/pii/S2213343720304553

* Some figures must be cited

*References must be updated.

Author Response

*The introduction does not tell the audience the problem statement's history and the tendency in this field.

DONE

The aim of this article is to summarise the 20 year experience of our research group, composed, at the beginning, of researchers from the Polytechnic University of Marche affiliated with the Clinic of infectious diromeases, the Clinic of Surgery and researchers from the Experimental Animal Models for Aging Unit of INRCA and from the Department of Inorganic Chemistry  of  Medical University of GdaÅ„sk(Poland). Our scientific production was initially focused on the antimicrobial properties of AMPs in medical device infection, biofilm and in sepsi.  With the arrival of researchers from dermatology, the institute of pathological anatomy and histology, our interest shifted to bacterial and fungal skin diseases. Our main goal was to provide new treatments to overcome multiresistant infections, particularly from Methicillin resistant staphylococcus aureus (MRSA) in chronic and surgical skin wounds, including burns. We studied the effect of AMPs not only for treating resistant bacteria related infection in skin but also for the wound healing.  

.”

* The recently published paper deal with MOFs and Heterocyclic compounds as  a new antimicrobial classes  not involved and must be cited:

https://link.springer.com/article/10.1007/s13738-022-02595-8

https://www.sciencedirect.com/science/article/abs/pii/S0022286021018536

https://www.sciencedirect.com/science/article/abs/pii/S101060302100441X

https://link.springer.com/article/10.1007/s10570-021-04147-4

https://www.sciencedirect.com/science/article/pii/S2405844020317990

https://www.sciencedirect.com/science/article/abs/pii/S2213343720304553

This article represents the history of the specific experience of our research group over the last 20 years, which is why many papers by other authors have not been included.

* Some figures must be cited

Done 

*References must be updated.

This article represents the history of the specific experience of our research group over the last 20 years, which is why many articles included were published up to 20 years ago.

Reviewer 2 Report

The review entitled «Antimicrobial peptides...», therefore, the main focus should be on AMPs, its properties, structure and prospects for use. However, in my view, the review so far looks like a set of brief descriptions of the studies that have been conducted. In addition, phrases such as "a lower bacterial load", "significant reduction", "substantial growth inhibition", "strongly affected", etc. require additional quantitative characteristics (doses, concentrations, percentages, etc.). Unfortunately, the first number appears only on page six of the review and the second on page eight.

Besides, it would be useful to give a brief description of the organization (or organizations) to which the research group belongs.

It is necessary to edit the English translation and depart from the often used laboratory slang, such as, for example, "Peptides in biofilm and medical devices" or "Peptides for fungi".

Main note: If the authors' task is to summarize the results of their investigations, if the authors consider their work as a single project that lasts 20 years, then it is necessary to clearly state the goals and objectives of the research, major achievements and future prospects.  It would be interesting to realize why they first worked with some objects and then with others, to understand why the change occurred (maybe it is due to the physical-chemical properties of AMPs, their sources, mechanism of action, etc.). It would be interesting to know whether the methods of testing and the laboratory models differ 20 years ago and now. Would any combination of AMP and antibiotics have synergistic effects, or are there definite rules? What is the mechanism of synergistic action? Is it possible to apply AMP parenterally or only locally, as a result of the study? What are the clinical perspectives of AMP?

Author Response

The review entitled «Antimicrobial peptides...», therefore, the main focus should be on AMPs, its properties, structure and prospects for use. However, in my view, the review so far looks like a set of brief descriptions of the studies that have been conducted.

We changed the title, Our Experience Over 20 Years: Antimicrobial Peptides Against Gram Positives, Gram Negatives and Fungi

In addition, phrases such as "a lower bacterial load", "significant reduction", "substantial growth inhibition", "strongly affected", etc. require additional quantitative characteristics (doses, concentrations, percentages, etc.). Unfortunately, the first number appears only on page six of the review and the second on page eight.

Done, see the corresponding paragraphs

Besides, it would be useful to give a brief description of the organization (or organizations) to which the research group belongs.

Our group is composed of researchers from the Polytechnic University of Marche affiliated with the Clinic of infectious diseases and the Clinic of dermatology.

It is necessary to edit the English translation and depart from the often used laboratory slang, such as, for example, "Peptides in biofilm and medical devices" or "Peptides for fungi".

Done, see heading of the paragraphs

Main note: If the authors' task is to summarize the results of their investigations, if the authors consider their work as a single project that lasts 20 years, then it is necessary to clearly state the goals and objectives of the research, major achievements and future prospects.  It would be interesting to realize why they first worked with some objects and then with others, to understand why the change occurred (maybe it is due to the physical-chemical properties of AMPs, their sources, mechanism of action, etc.). It would be interesting to know whether the methods of testing and the laboratory models differ 20 years ago and now. Would any combination of AMP and antibiotics have synergistic effects, or are there definite rules? What is the mechanism of synergistic action? Is it possible to apply AMP parenterally or only locally, as a result of the study? What are the clinical perspectives of AMP?

DONE

The aim of this article is to summarise the 20 year experience of our research group, composed, at the beginning, of researchers from the Polytechnic University of Marche affiliated with the Clinic of infectious diromeases, the Clinic of Surgery and researchers from the Experimental Animal Models for Aging Unit of INRCA and from the Department of Inorganic Chemistry  of  Medical University of GdaÅ„sk(Poland). Our scientific production was initially focused on the antimicrobial properties of AMPs in medical device infection, biofilm and in sepsi.  With the arrival of researchers from dermatology, the institute of pathological anatomy and histology, our interest shifted to bacterial and fungal skin diseases. Our main goal was to provide new treatments to overcome multiresistant infections, particularly from Methicillin resistant staphylococcus aureus (MRSA) in chronic and surgical skin wounds, including burns. We studied the effect of AMPs not only for treating resistant bacteria related infection in skin but also for the wound healing.”  

Round 2

Reviewer 2 Report

The manuscript has undergone considerable revision, and I suppose that it can be published in its present form after minor correction of English.